# Isotopic Characterization of Gaseous Mercury and Particulate Water-Soluble Organic Carbon Emitted from Open Grass Field Burning in Aso, Japan

**Satoshi Irei**

Department of Environment and Public Health, National Institute for Minamata Disease, 4058-18 Hama, Minamata, Kumamoto 867-0008, Japan; satoshi_irei@env.go.jp or satoshi.irei@gmail.com; Tel.: +81-966-63-3111 (ext. 760)

**Abstract:** Biomass burning is one of the major emitters of airborne particulate matter (PM) and gaseous mercury. In order to apply the isotopic fingerprinting method to process identification and source apportionment studies, isotopic characterizations of targeted substances at emission are indispensable. Here, we report the stable isotopic composition of total gaseous mercury (TGM) and the stable and radiocarbon isotopic composition of low-volatile water-soluble nitrogen (LV-WSN) and organic carbon (LV-WSOC) in PM emitted from open grass field burning in the Aso region of Japan. The measurement results showed that TGM concentrations in the air increased during the open field burning events, indicating the presence of TGM emissions. The results of LV-WSN analysis showed very low concentrations; therefore, the stable nitrogen isotope ratios could not be measured. The stable mercury isotope ratios exhibited lighter composition than those observed during non-biomass-burning days. The analysis of LV-WSOC revealed heavy stable carbon isotope ratios (average $\pm$ SD, $-18 \pm 2‰$), suggesting a substantial contribution from $C_4$ plant carbon. The $^{14}C$ analysis showed that more than 98% of the LV-WSOC was modern carbon, indicating the contribution of plant carbon to PM emitted from biomass burning. The findings here provide reference isotope compositions of TGM and particulate LV-WSOC from biomass burning in this region.

**Keywords:** $C_4$ plant; biomass burning; organic aerosol; stable mercury isotope; stable carbon isotope; radiocarbon isotope

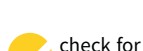



## 1. Introduction

Biomass burning, such as wildfires, biofuel burning, agricultural waste burning, and open field burning, is known to be among the major sources of atmospheric pollutants, including volatile species and airborne particulate matter (PM) [1–6]. The spatial distribution and chemical composition of airborne PM have attracted the attention of atmospheric scientists for the last two decades due to their relevance to cloud formation, which directly influences the Earth's radiative forcing, as well as to adverse health effects. It has also been reported that wildfires emit gaseous mercury species [7], which account for 8% of total global mercury emissions into the atmosphere [8]. Thus, studying biomass burning emissions can contribute to evaluating the effectiveness of the Minamata Convention on Mercury, which regulates the use of man-made mercury internationally in order to mitigate global mercury pollution. These subjects are intriguing for geoscientists in terms of material cycles in the biosphere. Despite the intensive studies completed on these issues to date, there are still many open questions remaining, such as the source apportionment and fate of atmospheric gaseous mercury and particulate organic carbon. The major difficulty is that the ambient air contains a complex mixture of those species from a variety of emission sources. Thus, concentration measurements alone are limited in providing convincing scientific evidence for their estimation in source contribution studies. To better trace the materials, isotopic compositions of elements, such as stable mercury isotope ratios (denoted as $\delta^x Hg$ hereafter), stable carbon isotope ratio ($\delta^{13}C$), stable nitrogen isotope ratio ($\delta^{15}N$),

and radiocarbon ($^{14}$C) isotope ratio, denoted as percent modern carbon (pMC) hereafter, have been drawing attention, since measurements of isotopic compositions increase the dimensions of the dataset, which allows us to objectively evaluate the mixing state and extent of processing [9–17]. To accomplish such a goal, knowledge of the initial isotopic compositions of those species at emission are indispensable [18–23]. To date, however, the reported initial stable isotopic compositions of emission sources are still limited.

The purpose of our research is to isotopically characterize total gaseous mercury (TGM), particulate low volatile water-soluble nitrogen (LV-WSN), and organic carbon (LV-WSOC) emitted from open field burning, called noyaki in Japanese. Specifically, the aim was to characterize five $\delta^x$Hg of TGM ($\delta^{199}$Hg, $\delta^{200}$Hg, $\delta^{201}$Hg, $\delta^{202}$Hg, and $\delta^{204}$Hg), $\delta^{15}$N of LV-WSN, and $\delta^{13}$C and pMC LV-WSOC in PM from noyaki in the Aso region. This is a continuation of studies reporting the measurement of TGM emitted from noyaki in 2019 [24]. With additional results from the same study conducted in 2021, the study of nitrogen and carbon in the water-soluble component extracted from airborne PM emitted from noyaki is newly presented.

## 2. Method and Materials

### 2.1. Open Grass Field Burning (Noyaki)

The Aso region is located in the center of Kyusyu Island in western Japan (Figure 1a,b). This region is made of a caldera, created by the volcanic activity of Mt. Aso. Grass fields are found in the mountainside of Mt. Aso and the outer edge of the caldera. The grass fields in this region have been burned every spring since the 8th–12th century, originally for the purpose of maintaining a clear sight for hunting wild animals, but nowadays for the purpose of protecting the grass fields from wild forestation. Noyaki in Aso is the largest open field burning event in Japan. According to a plant survey by Aso-Kujyu National Park [25], most grass habitats are "susuki" (*Miscanthus sinensis*) and "nezasa" (*Pleioblastus chino* var. *viridis*) (Figure 2). The former is identified as a C$_4$ plant [26] and is a typical wild plant found everywhere in Japan, while the latter is found in limited areas, but is the dominant plant species found in the Aso region [27]. Nezasa is assumed to be a C$_3$ plant due to the name "sasa" indicating a family of bamboo, which is a C$_3$ plant [28]. Furthermore, the shape of the leaves resembles that of bamboo leaves and the leaf venation is green, which is an indication of chloroplast in the venation, a feature of C$_4$ plants [29]. To the best of our knowledge, there is no solid scientific evidence to identify it as C$_3$ or C$_4$.

### 2.2. Sampling

Samples collected in the series of noyaki studies were summarized in Supplementary Table S1. Noyaki sampling was carried out at two locations, Daikanpou and Namino, in spring 2019, as well as in three locations, Komezuka, Ubuyama, and Kitayama, in spring 2021 (Figure 1c). TGM and PM emitted from noyaki were sampled from the car window (Figure 3). The car was moved and stopped repeatedly in the downstream of the noyaki plume so that the emissions could be directly captured while a safe distance from the fire could be maintained. The sampling device consisted of an open-face filter holder (Innovation nilu AS, Kjeller, Norway) with a 47 mm o.d. polytetrafluoroethylene (PTFE) coated glass filter (Pallflex, Emfab, Pall Corp., Port Washington, NY, USA), 1/2 inch o.d. × ~60 cm long perfluoroalkoxy alkane (PFA) tubing (Tombo 9003-PFA, Nichias Corp., Tokyo, Japan), a newly developed big gold mercury trap (BAuT) (Irei et al., 2020), a float flow meter (RK230, KOFLOC, Kyoto, Japan), and a sampling diaphragm pump (N860FTE, KNF, Freiburg, Germany). PTFE connectors for the BAuT were specially manufactured (COSMOS VID, Fukuoka, Japan), and other standard size connectors used were either PFA (Swagelok, Solon, OH, USA) or PTFE (Flowell 30 series, Flowell Corp., Yokohama, Japan) connectors. An uninterrupted power supply (SURTA1500XLJ, American Power Conservation, West Kingston, RI, USA) was used as the power source for the diaphragm pump. TGM samples collected using the BAuT setup were summarized in Supplementary Table S2.

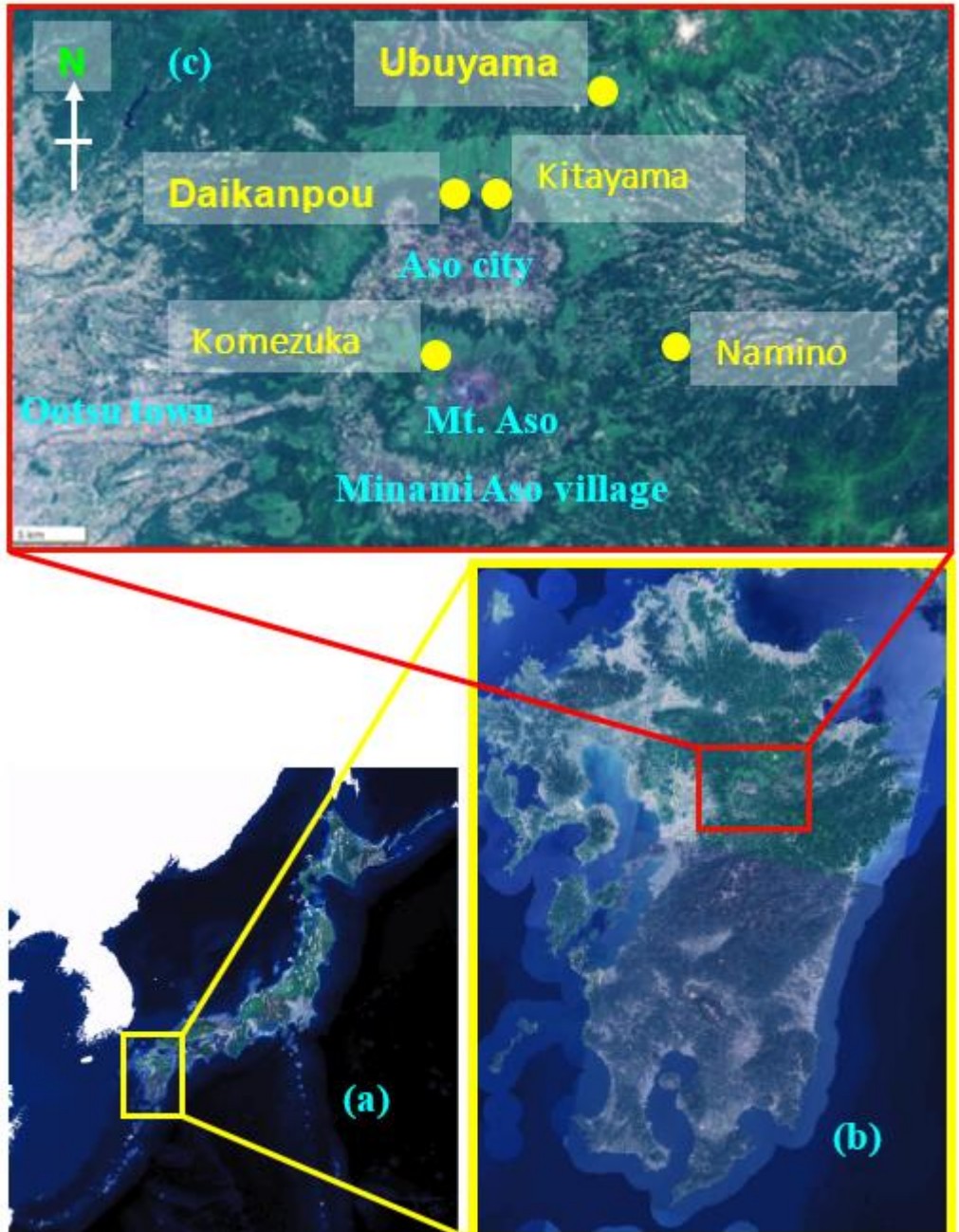

**Figure 1.** Map of (**a**) Japan, (**b**) Kyusyu, and (**c**) Aso region with sampling locations (yellow circles). Courtesy of Geospatial Information Authority of Japan, which permitted the use of GSI maps.

For the noyaki sampling conducted at Ubuyama, Kitayama in 2021, as well as all background air sampling conducted in 2021, double BAuT sampling (TGM sampling using two BAuTs connected in series; the setup is shown in Figure 3) was conducted to monitor breakthrough TGM from the front BAuT. TGM in the air was also sampled using commercially available gold-coated sand traps (4 mm × 160 mm; Nippon Instruments Corp., Osaka, Japan) with a mini pump (MP-W5P, SIBATA, Scientific Technology Ltd., Souka, Japan) to compare BAuT and regular TGM sampling. It should be noted that this comparison was made only for the study completed in 2021, due to the sampling and measurement failure of samples collected in 2019. It should also be noted that the PTFE filter was replaced at a certain point during the BAuT sampling, as it was heavily loaded with PM. In addition, stand-alone filter sampling was occasionally conducted separately from the BAuT sampling in order to collect more filter samples.

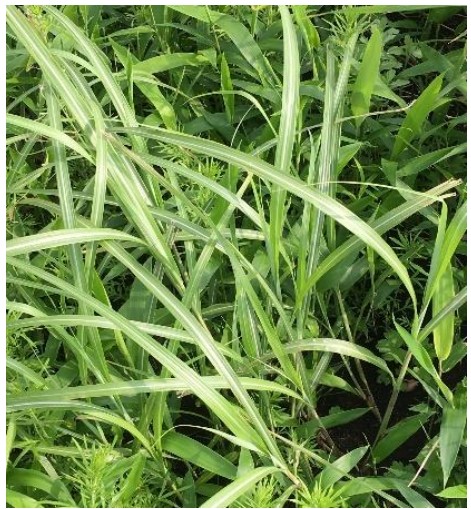
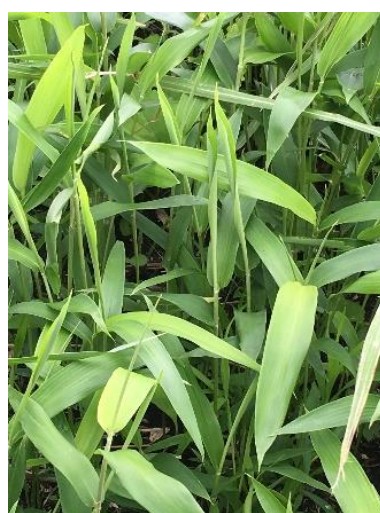

**Figure 2.** Photos of "susuki" (*Miscanthus sinensis*, **left**) and "nezasa" (*Pleioblastus chino* var. *viridis*, **right**).

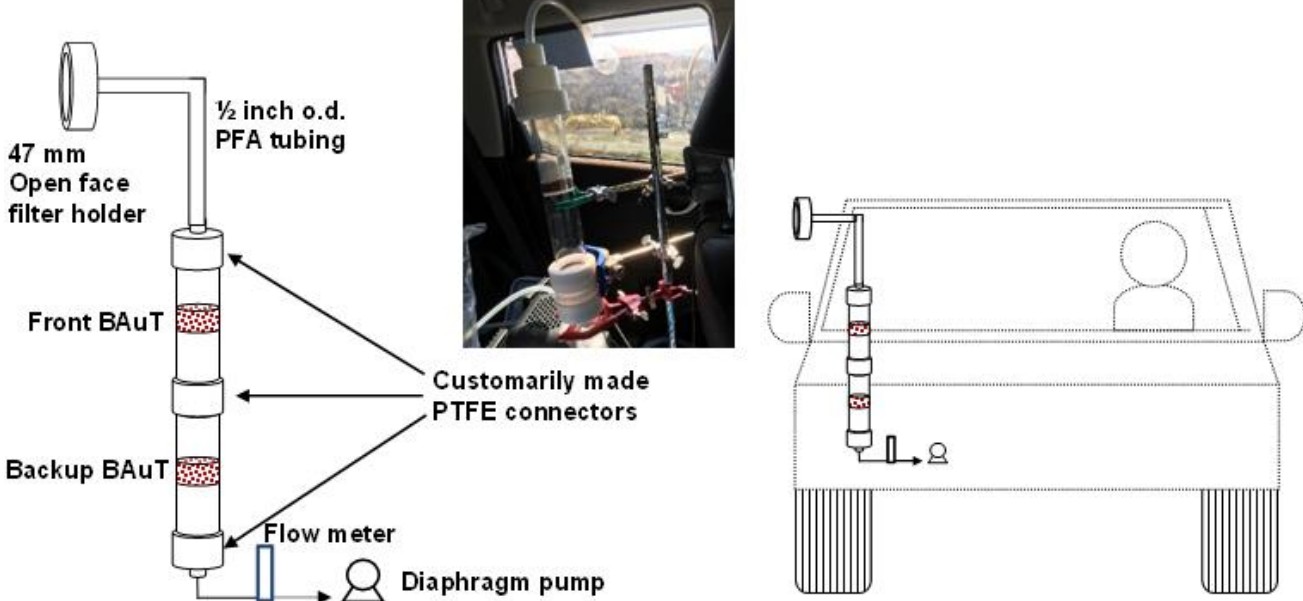

**Figure 3.** Sampling setup for GEM and PM from Aso open grass field burning.

Noyaki sampling was successfully carried out twice in spring 2019 and three times in 2021. Overall, 2 BAuT samples for TGM and 10 filter samples for water-soluble particulate species were collected during 2019, and 3 BAuT samples and 5 filter samples were collected during 2021 (Supplementary Tables S2 and S3). In addition to the noyaki samples, background air sampling was performed to see the impact from noyaki. Two BAuT background air samples and two background filter samples were collected in 2019, and two BAuT samples and five filter samples were collected in 2021. In addition, *Miscanthus sinensis* and *Pleioblastus chino* var. *viridis* samples were collected from the waste of mown grass in 2021 for $\delta^{13}$C analysis.

### 2.3. TGM Analysis

The analysis of trapped TGM by BAuT is described elsewhere [24]. Briefly, BAuT samples, including front and backup samples, were brought to the laboratory, and the captured TGM in each BAuT was pre-concentrated in a commercially available gold-coated sand trap. The pre-concentration was made by heating the BAuT to 873 K for 1 h under 0.5 mL min$^{-1}$ flow of mercury-free dried air, then the released gaseous elemental mercury

(GEM) was trapped again using the gold trap. The pre-concentration step was necessary for isotope analysis so that nearly all the captured TGM in the front BAuT was efficiently transferred into a 2L Tedler bag, and the TGM sampled using the conventional mercury trap and the mini-pump was analyzed for atmospheric concentration by a cold vapor–atomic fluorescent spectrometer (WA-5F, Nippon Instruments Corp.). This pre-concentration step was taken for the backup BAuT samples as well. The pre-concentrated mercury sample from the front BAuT was attached to a 2L Tedler bag with a single stopcock (AS ONE, Osaka, Japan), then heated to 873 K again to transfer the mercury into the bag. Prior to the transfer, 5 mL trapping acid solution (40% reversed aqua regia) was pre-loaded in the Tedler bag. After the transfer, the bag was left for a week to convert GEM to $Hg^{2+}$ in the solution, which was validated elsewhere [24]. The acid sample solution was then subjected to isotope measurement. Stable mercury isotope ratios were measured by a cold vapor generator (HGX-200, Teledyne CETAC Technologies, Omaha, NE, USA) coupled with a multicollector inductively coupled plasma mass spectrometer (MC-ICP-MS; Neptune Plus, Thermo Fisher Scientific GmbH, Bremen, Germany). Measurement artifacts (unfavorable isotope fractionations) occurring in ICP sections were corrected by analyzing the simultaneously introduced SRM 997 thallium aerosol produced by an aerosol generator (Aridus II, Teledyne CETAC Technologies). The delta expression of the stable mercury isotope ratio is defined as follows:

$$\delta^x Hg \ (\text{‰}) = \left[ \frac{\left( \frac{{}^x Hg}{{}^{198} Hg} \right)_{sample}}{\left( \frac{{}^x Hg}{{}^{198} Hg} \right)_{3133}} - 1 \right] \times 1000 \tag{1}$$

where x stands for the stable mercury isotope with mass x, and the bracketed isotope ratios with subscripts "sample" and "3133" indicate the stable mercury isotope ratios of mass x relative to mass 198 for the sample and SRM 3133 (NIST), respectively.

Overall recovery and isotope measurement tests were performed by introducing the GEM standard from the NIST 8610 standard reference material to the double BAuT setup (Supplementary Figure S1). The recovery tests were conducted with a carrier gas of mercury-free dried air during the GEM transfer from the BAuT to the conventional trap. Further recovery tests were conducted using GEM from a standard box (MB-1, Nippon Instruments Corp.) to find the difference in recovery yields using either mercury-free dried air or pure nitrogen (Fukuoka Sanso) during the GEM transfer.

Pre-concentrated TGM from backup BAuTs and TGM samples collected with the conventional gold traps were quantitatively analyzed by the CV-AFS. The detection limit was on the order of sub-picograms, and the typical precision based on its calibration was 3% or better.

### 2.4. Filter Sample Analysis

Collected PM filter samples were analyzed for nitrogen and carbon among the low-volatile water-soluble organic components in PM from noyaki. The details of the analytical procedure were described elsewhere [30]. Briefly, filter samples were submerged in 15 mL Milli-Q water (Milli-Q Integral 3, Merck KGaA, Darmstadt, Germany) and then sonicated for 15 min. The water-soluble component was transferred to a 100 mL evaporation flask (COSMOS VID). The sonication procedure was repeated twice with an additional 10 mL of Milli-Q water. All extracts were combined in the evaporating flask. The volume was reduced to approximately 0.2 mL, then transferred to a conically shaped microbial (GL Science, Osaka, Japan). Further volume reduction to 0.2 mL was completely by gently blowing the extract in the vial (GL Sciences) with pure nitrogen gas. A certain volume of each concentrated extract was then pipetted into a tin cup (Lüdi Swiss AG, Flawil, Switzerland), together with a drop of 0.1 M hydrochloric acid solution, then the sample solution was dried under a gentle stream of pure nitrogen to produce residual low-volatile water-

soluble nitrogen (LV-WSN) and organic carbon (LV-WSOC). This extract was subjected to quantitative and isotope analyses.

Quantities of nitrogen and carbon in LV-WSN and LV-WSOC and $\delta^{13}$C of LV-WSOC were analyzed by an elemental analyzer (EA; Flush EA 1112, Thermo Fisher Scientific GmbH) coupled with an isotope ratio mass spectrometer (IRMS; Delta Plus, Thermo Fisher Scientific GmbH). The stable carbon isotope ratio in delta expression for $\delta^{13}$C is defined as follows:

$$\delta^{13}\text{C}\ (\text{‰}) = \left[ \frac{\left(\frac{^{13}\text{C}}{^{12}\text{C}}\right)_{\text{sample}}}{\left(\frac{^{13}\text{C}}{^{12}\text{C}}\right)_{\text{VPDB}}} - 1 \right] \times 1000 \tag{2}$$

where the bracketed isotope ratios with subscripts "sample" and "VPDB" indicate the stable carbon isotope ratios for the sample and the reference Vienna Pee Dee Belemnite, respectively. The minimum quantities for stable nitrogen and carbon isotope analysis were 50 μgN and 20 μgC, respectively. For radiocarbon ($^{14}$C) analysis, LV-WSOC extracts sampled on the same day were combined to have sufficient carbon. The LV-WSOC samples were converted to graphite, and the ratio of $^{14}$C to $^{12}$C was analyzed using an acceleration mass spectrometer (AMS; Pelletron 1.5SDH-1, National Electrostatic Corp., Middleton, WI, USA). The overall minimum carbon quantity required for $^{14}$C analysis was 300 μgC. Here, the $^{14}$C ratio ($^{14}$C/$^{12}$C) of the samples was relative to that of SRM 4990c (NIST, Gaithersburg, MD, USA) and was reported in percent modern carbon (pMC) with and without $\delta^{13}$C corrections.

Naturally dried plant samples were cut into small pieces, vacuum dried, and ground to powder using a multi-bead shocker (Yasui Kikai, Osaka, Japan). Then 0.1–0.2 mg of the powder was put into a tin cup (Lüdy Swiss AG) and analyzed by EA-IRMS for $\delta^{13}$C.

## 3. Results and Discussion

### 3.1. Blank and Method Validations

The results of the analytical tests for TGM isotope measurements using gaseous elemental mercury prepared from SRM 8610 showed recovery yields of $70 \pm 3\%$, $65 \pm 7\%$, and $90 \pm 8\%$ for 5.2, 15.5, and 41.3 ng mercury introduced, respectively (Supplementary Table S4). The $\delta^{\text{x}}$Hg measurements of these samples for all the isotope ratios, except $\delta^{199}$Hg, showed significant differences between the measured and reference $\delta^{\text{x}}$Hg, and the average differences ($\pm 2 \times$ propagated uncertainty) were smallest for the test with 41.3 ng mercury, from $-0.16 \pm 0.09$‰ to $-0.37 \pm 0.22$‰, while the differences for 5.2 ng and 15.5 ng were in comparable level, from $0.24 \pm 0.20$‰ to $0.90 \pm 0.50$‰ (Supplementary Table S4). For $\delta^{199}$Hg there was not a significant difference. The significant differences may have been due to unfavorable small fractionations for the heavy isotopes occurring during the TGM transfer from BAuTs to conventional gold traps, since we used air for the transfer. The results of the mercury transfer tests from BAuTs to conventional traps using either mercury-free air or pure nitrogen as the carrier gas showed recovery yields of $68 \pm 5\%$ ($n = 3$) and $96\%$ ($n = 1$), respectively. Thus, we speculate that the use of air as carrier gas caused an unfavorable reaction during the transfer. Unfortunately, dried mercury-free air was used to transfer TGM samples from the noyaki studies. Therefore, our $\delta^{\text{x}}$Hg data may have positive biases that needs to be corrected for $\delta^{200}$Hg, $\delta^{201}$Hg, $\delta^{202}$Hg, and $\delta^{204}$Hg when the captured TGM was between 5.2 and 15.5 ng. For the mass 41 g the biases are very small, slightly above $3\sigma$ for those isotope ratios.

The analysis of backup BAuTs, which contained the breakthrough TGM from the front BAuTs, showed TGM of 100 pg or less. The minimum quantity of TGM captured by the front BAuT during noyaki and the background air sampling was 5 ng, indicating the breakthrough TGM was 2% or less.

The analysis of eight blank filters resulted in nitrogen blank values lower than 1 μgN per filter. Due to the limited number of extracts, the stable nitrogen isotope ratio was not measured. The analysis of blank carbon showed an average blank value of 10 μgC per filter with $\delta^{13}$C of $-23.9$‰. Further, the $^{14}$C composition, or pMC, for the blank carbon was not

measured due to the limited number of blank extracts, which were used for the $\delta^{13}C$ analysis. The analysis of carbon mass and $\delta^{13}C$ was conducted using the IAEA-C6 sucrose standard, which has $\delta^{13}C$ of $-10.8 \pm 0.5‰$ ($\pm$uncertainty); see Supplementary Subsection S.2. for this evaluation. Briefly, within the range of carbon masses found in our LV-WSOC samples, the impact of the blank value was significant. Thus, the blank correction was needed for the yield of carbon masses and their $\delta^{13}C$ values. The blank-corrected $\delta^{13}C$ value agreed with the reference $\delta^{13}C$ value within the propagated uncertainty (0.51 to 0.78‰, depending on the spiked carbon mass). Due to the unknown pMC in the blank carbon, the blank correction was not made for $^{14}C$ data. Based on the blank value referred to previously, the blank carbon in the combined filter extracts for $^{14}C$ analysis accounted for 5% in the Daikanpou sample, 2% in the Namino sample, 1% in the Ubuyama sample, and 4% in the Kitayama sample. Considering the blank carbon as the dead carbon (assuming the petroleum originated organic material used in the production of PTFE-coated filters as the major source of blank carbon) and our sample carbon as modern carbon, the blank fractions would result in a maximum 1–5% bias in the pMC estimation. The expected maximum impact was still small and the actual blank pMC was unknown; therefore, pMC values reported here were not corrected for the blank.

*3.2. TGM*

The measurement results showed that except for the sample collected on 6 April 2019, significant amounts of TGM were emitted from the noyaki events (Table 1). A comparison of the TGM concentrations with the background air revealed that the atmospheric TGM concentrations were enhanced by approximately four times. It is worth noting that the amount of TGM collected on 10 May 2021, was more than double that collected from the background air on different days. The arrival of Kosa, desert dust from China, was recorded in the Kyusyu region, therefore, this highly loaded TGM was the transboundary TGM. As discussed in the previous subsection, however, all the TGM concentrations determined by BAuT sampling may have been lower than the actual concentrations, due to the artifact from the transfer of TGM from BAuT to a conventional trap. A comparison between the concentrations determined by BAuT and conventional trap sampling made in the spring of 2021 showed lower concentrations determined by the former (Supplementary Table S6). Thus, the corrections on $\delta^{x}Hg$ values for the biases caused by this analytical artifact were considered (see below).

**Table 1.** Results of $\delta^{x}Hg$ of TGM emitted from noyaki.

| Scheme 199 | Total Hg Sampled | Conc. in air | $\delta^{199}Hg$ | $\delta^{200}Hg$ | $\delta^{201}Hg$ | $\delta^{202}Hg$ | $\delta^{204}Hg$ |
|---|---|---|---|---|---|---|---|
| | ng | ng m$^{-3}$ | | | ‰ | | |
| *Noyaki samples* | | | | | | | |
| 24 March 2019, Daikanpou BAuT * | 14.5 | 3.7 | −0.58 | −0.58 | −1.01 | −0.93 | −1.40 |
| 6 April 2019, Namino BAuT * | 5.0 | 1.4 | −1.11 | −0.69 | −1.23 | −1.37 | 1.18 |
| 23 March 2021, Ubuyama BAuT | 38.0 | 4.0 | −0.60 | −0.66 | −1.40 | −1.34 | −1.99 |
| 27 March 2021, Kitayama BAuT | 43.7 | 4.3 | −0.54 | −0.57 | −1.01 | −1.03 | −1.43 |
| *Background air samples* | | | | | | | |
| 31 March 2019, Namino BAuT * | 5.7 | 0.8 | −0.05 | 0.44 | 0.35 | 1.06 | 1.30 |
| 23 May 2019, Namino BAuT * | 15.0 | 0.7 | −0.02 | 0.13 | −0.06 | 0.16 | 0.49 |
| 19 April 2021, Komezuka BAuT | 17.0 | 1.0 | −0.29 | −0.15 | −0.12 | −0.11 | −0.24 |
| 23 April 2021, Ubuyama BAuT | 18.7 | 1.0 | −0.34 | −0.27 | −0.11 | 0.03 | −0.17 |
| 10 May 2021, Kitayama BAuT | 38.0 | 1.3 | −0.21 | −0.18 | −0.29 | −0.28 | −−0.32 |

* Published data [24].

Given the extent of the isotope fractionations observed in the overall recovery tests accompanying the Hg loss shown in Supplementary Table S6, the $\delta^x$Hg values were corrected for the artifact fractionations (Table 2). Here, corrections for the $\delta^x$Hg measurement biases observed in the recovery tests with 5.2–15.5 ng Hg (30% Hg loss) were applied to all the noyaki and background air data since the losses of Hg shown in Table S6 were larger than the loss found in the recovery tests with 41 ng Hg, and the sample size of noyaki and background air samples collected in 2019 were overall smaller than the samples collected in 2021.

**Table 2.** Biases corrected $\delta^x$Hg.

| Sample | $\delta^{199}$Hg | $\delta^{200}$Hg | $\delta^{201}$Hg | $\delta^{202}$Hg | $\delta^{204}$Hg |
|---|---|---|---|---|---|
| | | | ‰ | | |
| *Noyaki samples* | | | | | |
| 24 March 2019, Daikanpou BAuT | −0.66 | −0.91 | −1.48 | −1.62 | −2.29 |
| 2019 April 6, Namino BAuT | −1.19 | −1.02 | −1.70 | −2.06 | 0.29 |
| 23 March 2021, Ubuyama BAuT | −0.68 | −0.99 | −1.87 | −2.03 | −2.88 |
| 27 March 2021, Kitayama BAuT | −0.62 | −0.90 | −1.48 | −1.72 | −2.32 |
| *Background air samples* | | | | | |
| 31 March 2019, Namino BAuT | −0.13 | 0.11 | −0.12 | 0.37 | 0.41 |
| 23 May 2019, Namino BAuT | −0.10 | −0.20 | −0.53 | −0.53 | −0.40 |
| 19 April 2021, Komezuka BAuT | −0.37 | −0.48 | −0.59 | −0.80 | −1.13 |
| 23 April 2021, Ubuyama BAuT | −0.42 | −0.60 | −0.58 | −0.66 | −1.06 |
| 10 May 2021, Kitayama BAuT | −0.29 | −−0.51 | −0.76 | −0.97 | −1.21 |

$\delta^x$Hg values displayed that except for the $\delta^{204}$Hg value of the 6 April 2019, Namino sample, the $\delta^x$Hg values of TGM samples emitted from noyaki were significantly smaller than those from the background air (Table 2). According to the reports for $\delta^x$Hg values of Hg in plants and soils, low $\delta^x$Hg values (or light isotopic compositions) have been found [31–35]. Our observations were consistent with their reports. For further analysis, the extent of mass independent fractionation (MIF) was evaluated.

MIF can be evaluated by the deviation of the observed $\delta^x$Hg from the predicted $\delta^x$Hg using $\delta^{202}$Hg [36]:

$$\Delta^x\text{Hg (‰)} \approx \delta^x\text{Hg} - \left(\delta^{202}\text{Hg} \times \beta_x\right) \tag{3}$$

where $\beta_x$ is the mass dependent fractionation factor for mass x (0.2520, 0.5024, 0.7520, and 1.4930 for mass 199, 200, 201, and 204, respectively). A scatter plot of $\delta^{202}$Hg against $\Delta^{199}$Hg showed negative values for both (Figure 4), which was a common feature of observed for Hg in plants [31,33–35] and in soil [32,35]. However, a plot of $\Delta^{201}$Hg versus $\Delta^{199}$Hg demonstrated scattered plots, regardless of the biases were corrected (Figure 5), even though reported plots display high correlations. Particularly, the noyaki sample collected on 6 April 2019 in Namino showed extreme $\Delta^x$Hg values. This sample also showed unusual values in a plot of $\Delta^{200}$Hg versus $\Delta^{204}$Hg, and we do not know if these extreme $\delta^x$Hg values were caused by the combustion. Even though this study could not distinguish the origins of Hg between plant and soil, the deposited mercury on the surface over a year still showed $\delta^x$Hg values consistent with those observed by others.

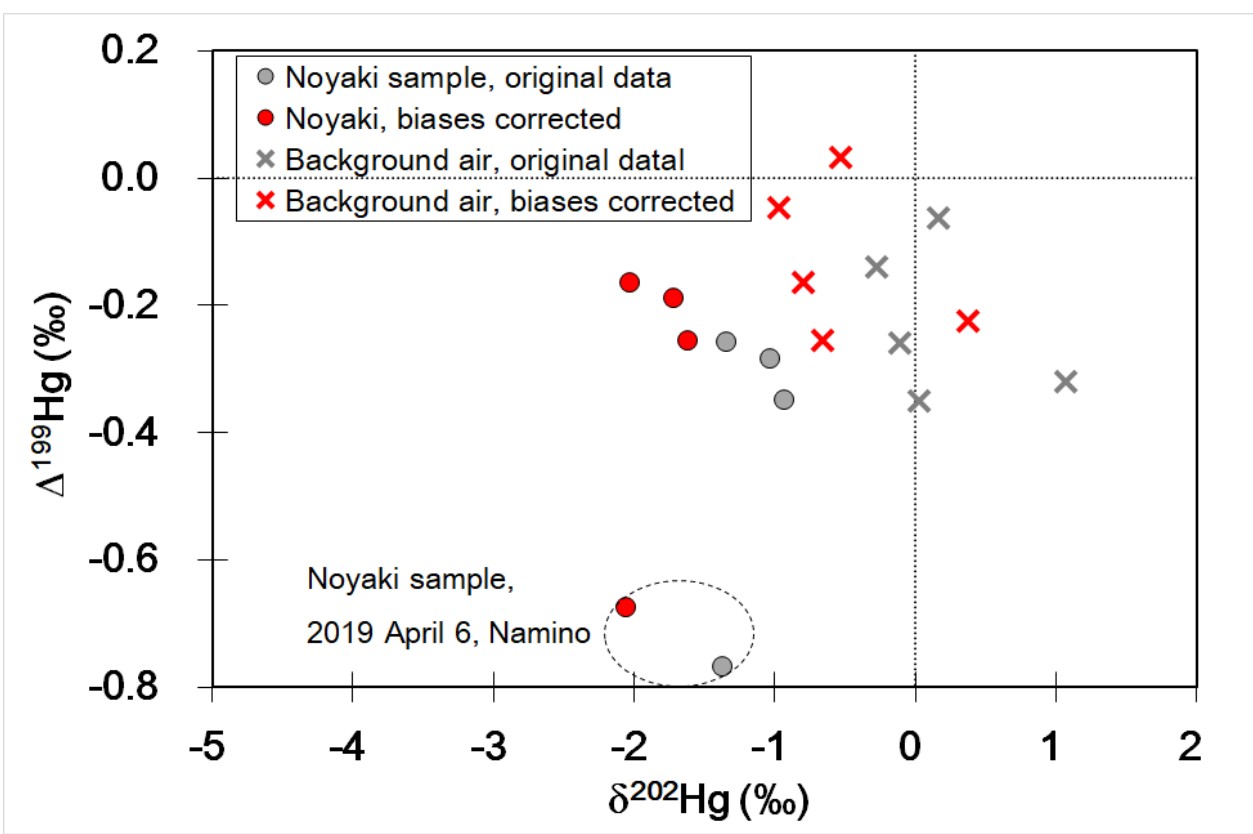

**Figure 4.** Scatter plot of $\Delta^{199}$Hg versus $\delta^{202}$Hg for TGM from noyaki and background air samples before and after the corrections for the measurement biases were made.

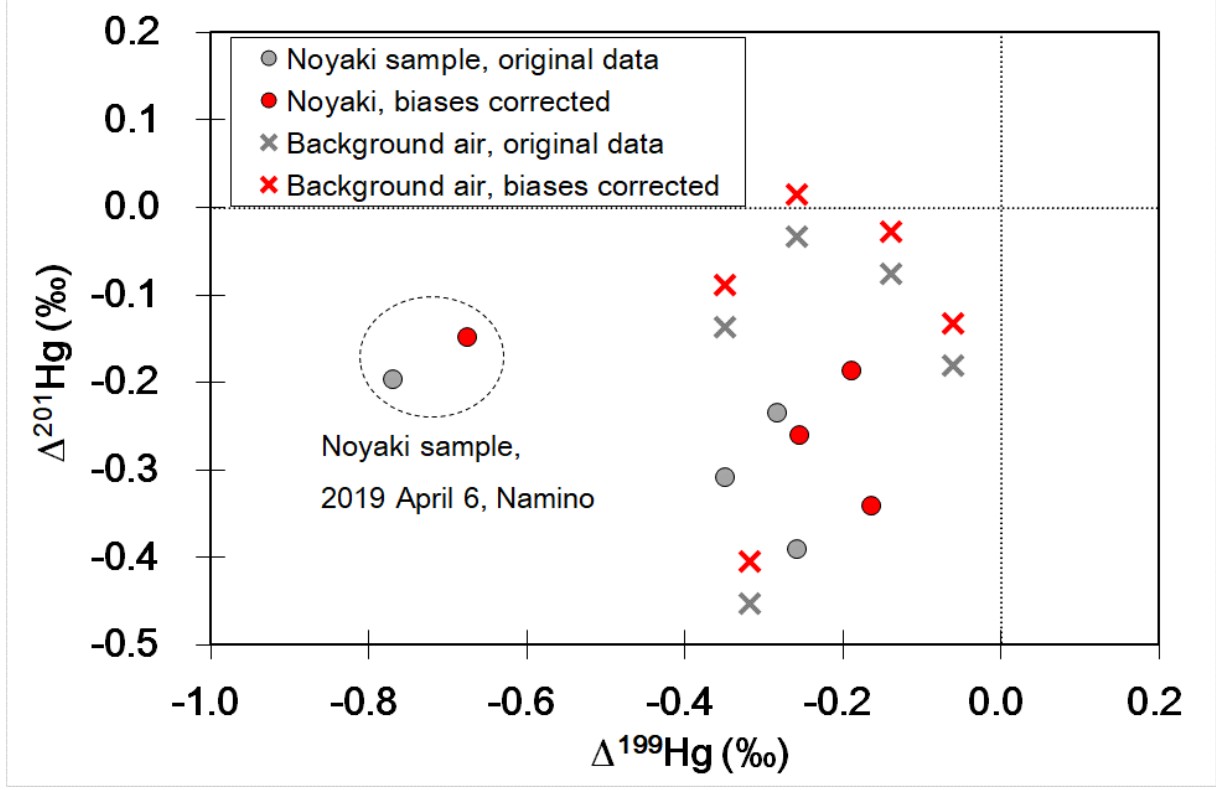

**Figure 5.** Scatter plot of $\Delta^{199}$Hg versus $\Delta^{201}$Hg for TGM from noyaki and background air samples before and after the corrections for the measurement biases were made.

### 3.3. LV-WSN in PM from Noyaki

The results of LV-WSN analysis of PM from noyaki showed low nitrogen content, implying a minor contribution of LV-WSN in PM emitted from biomass burning (Table 3). Some additional tin cup noyaki samples were prepared without spikes of HCl solution, the carbonate remover, to see if the low nitrogen concentrations were artifacts caused by spikes of 0.1 M hydrochloric acid. We observed no significant difference between samples with and without HC spikes; therefore, the observation of the low particulate LV-WSN concentrations was conclusive. Overall, the observed atmospheric concentration range of most of the particulate LV-WSN samples were from below the detection limit to 22 $\mu$gN m$^{-3}$. These quantities were not sufficient to determine their $\delta^{15}$N values; sampling larger air volume is needed for this determination.

**Table 3.** Nitrogen mass, carbon mass, and $\delta^{13}$C of low-volatile water-soluble organic component in PM emitted from Noyaki.

| Sample | Nitrogen | | Carbon | | |
|---|---|---|---|---|---|
| | Mass * | Conc. * | Mass * | Conc. * | $\delta^{13}$C$_{VPDB}$ * |
| | $\mu$gN filter$^{-1}$ | $\mu$gN m$^{-3}$ | $\mu$gC filter$^{-1}$ | $\mu$gC m$^{-3}$ | ‰ |
| *Noyaki samples* | | | | | |
| 24 March 2019, Daikanpo 1st BAuT filter 1 | LDL $^\dagger$ | LDL $^\dagger$ | 102 | 37 | −14.43 |
| 24 March 2019, Daikanpo 1st BAuT filter 2 | 2 | 0.6 | 128 | 36 | −18.36 |
| 24 March 2019, Daikanpo 1st BAuT filter 3 | 5 | 1.3 | 584 | 140 | −18.96 |
| 24 March 2019, Daikanpo 2nd BAuT filter 2 | LDL $^\dagger$ | LDL $^\dagger$ | 85 | 28 | −20.89 |
| 24 March 2019, Daikanpo filter 1 | 3 | 2.5 | 71 | 53 | −13.81 |
| 24 March 2019, Daikanpo filter 2 | 0 | 0.2 | 68 | 40 | −17.39 |
| 24 March 2019, Daikanpo filter 3 | 23 | 11.6 | 396 | 203 | −19.03 |
| 24 March 2019, Daikanpo filter 4 | 1 | 1.0 | 69 | 47 | −20.38 |
| 6 April 2019, Namino BAuT filter | 56 | 16.0 | 472 | 135 | −19.36 |
| 6 April 2019, Namino noyaki filter | 36 | 21.1 | 323 | 191 | −19.11 |
| 14 March 2021, Komezuka BAuT filter | 2 | 0.2 | 127 | 11 | −14.10 |
| 23 March 2021, Ubuyama BAuT filter | 84 | 8.7 | 1018 | 106 | −21.65 |
| 27 March 2021, Kitayama BAuT filter 1 | LDL $^\dagger$ | LDL $^\dagger$ | 202 | 38 | −17.56 |
| 27 March 2021, Kitayama BAuT filter 2 | 4 | 1.8 | 517 | 243 | −18.94 |
| 27 March 2021, Kitayama BAuT filter 3 | LDL $^\dagger$ | LDL $^\dagger$ | 107 | 39 | −16.26 |
| *Background air samples* | | | | | |
| 31 March 2019, Namino background | 3 | 0.5 | 7 | 1 | 1.90 |
| 23 May 2019, Namino background | NA $^\ddagger$ | NA $^\ddagger$ | NA $^\ddagger$ | NaA$^\ddagger$ | Na $^\ddagger$ |
| 23 April 2021, Ubuyama background | 9 | 0.5 | 18 | 1 | −9.34 |
| 10 May 2021, Kitayama background | 49 | 1.7 | 32 | 1 | −13.90 |
| 25 May 2021, Kitayama background | 11 | 0.4 | 25 | 1 | −13.04 |
| 1 June 2021, Namino background | 7 | 0.3 | 2 | 0.1 | 14.31 |

\* Values were blank-corrected. $^\dagger$ Lower than detection limit. $^\ddagger$ Analysis failed and data are not available.

### 3.4. $\delta^{13}$C of Habitat Plants and LV-WSOC in PM from Noyaki

The results of plant carbon analysis showed $\delta^{13}$C values of −11.97 ± 0.13‰ and −30.25 ± 0.05‰ for *Miscanthus sinensis* and *Pleioblastus chino* var. *viridis*, respectively. This provided evidence supporting that those plants were C$_4$ and C$_3$ plants, respectively.

The results of particulate LV-WSOC analysis showed variation from −13.8 to −21.7‰, mostly above −20‰ (Table 3). It should be noted that even though $\delta^{13}$C values for the background air indicated heavier isotopic compositions compared to the noyaki samples, those values were unreliable since total carbon loaded on the filters was very low and the impact of the blank value was significant. An improved sampling method was needed for the determination of $\delta^{13}$C with background LV-WSOC. For the noyaki samples, the average

$\delta^{13}$C value was $-18 \pm 2$‰. Using the plant $\delta^{13}$C values referred to earlier in a mass balance, the contribution of LV-WSOC from *Miscanthus sinensis* and *Pleioblastus chino* var. *viridis* was estimated to be 67 and 33%, respectively, implying the fractions of the habitats of those plants in this region. This isotopic composition was heavier than those typically observed in the East Asian region, and it was attributed to the contribution of carbon from *Miscanthus sinensis*, a $C_4$ plant. It had been reported that the air in the Kyusyu/Okinawa region, where air from China flows into, had background LV-WSOC with $\delta^{13}$C of $-15$‰ [30], implying the influence of $C_4$ plant combustion in neighboring countries, as well as the possibility of burning of agricultural waste from corn fields. The reason was that substantial amounts of open biomass burning was prohibited by law in Japan in general, and types of $C_4$ plants were thought to be rare. Similar background $\delta^{13}$C values were still observed in recent studies [37]. Our study, however, demonstrated that $C_4$ plants, the source of such high $\delta^{13}$C values, exist in our surroundings, and the small amount of burning of yard and agricultural wastes was trivial. Thus, local biomass burning can be a source of background LV-WSOC with high $^{13}$C content.

### 3.5. Radiocarbon of LV-WSOC in PM

In $^{14}$C analysis it is customary to correct $^{14}$C isotope ratios using the measured $\delta^{13}$C values as the reference values [38]. This is to define the starting point of $^{14}$C aging. The correction assumes that the deviation from $-25$‰ is evidence of bias in $^{14}$C aging. However, this assumption is not valid in some cases [39], and in our case as well, since our purpose for using pMC is not about aging but understanding the mixing state of modern and dead carbons. It is also since one of the expected end members was an annually growing $C_4$ plant species, which has $\delta^{13}$C values significantly heavier than $-25$‰, as discussed in the previous subsection. Corrected and uncorrected pMCs were presented in Table 4, together with the $\delta^{13}$C measured by AMS. The difference between corrected and uncorrected pMCs ranged from 0.5 to 1.7%. Such differences may occasionally be significant in $^{14}$C aging but have a minor impact on source apportionment.

**Table 4.** $\delta^{13}$C and pMC of LV-WSOC found in PM collected from Aso open grass field burning.

| Sample | $\delta^{13}$C | pMC [†] | pMC [‡] |
|---|---|---|---|
| | ‰ | % | % |
| 24 March 2019, Daikanpou | $-17.2 \pm 0.2$ | $101.2 \pm 0.3$ | $99.5 \pm 0.3$ |
| 6 April 2019, Namino | $-16.8 \pm 0.2$ | $99.8 \pm 0.3$ | $98.1 \pm 0.3$ |
| 27 March 2021, Kitayama | $-22.4 \pm 0.2$ | $100.1 \pm 0.3$ | $99.6 \pm 0.3$ |
| 23 March 2021, Ubuyama | $-18.8 \pm 0.3$ | $99.4 \pm 0.3$ | $98.2 \pm 0.3$ |

[†] Before $\delta^{13}$C correction. [‡] After $\delta^{13}$C correction.

The results of the radiocarbon analysis of LV-WSOC in PM from noyaki showed pMC higher than 99 and 98% with and without $\delta^{13}$C corrections, respectively (Table 4). The observations confirmed no substantial $^{14}$C isotope fractionation during the biomass burning of habitat plants. It was often assumed that the radiocarbon composition emitted from biomass burning would reflect that of burnt material. In this study we confirmed this, showing that noyaki did not significantly discriminate the radiocarbon of LV-WSOC and was a useful tool in source apportionment of atmospheric particulate LV-WSOC.

### 4. Conclusions

This study revealed that $\delta^x$Hg values of TGM emitted from noyaki indicated light compositions, in agreement with the reported $\delta^x$Hg values of mercury found in plants and soil. This feature can potentially fingerprint atmospheric TGM from biomass burning. The analysis of nitrogen content in the LV-WS component showed very small amounts, and noyaki was likely a very minor source of nitrogen found in the water-soluble component of airborne PM. The analysis of carbon content showed substantial amounts of carbon in

the LV-WSOC of PM and a $\delta^{13}$C of $-18‰$, on average. A comparison of $\delta^{13}$C values of local plants revealed that 67% of carbon originated from "susuki" (*Miscanthus sinensis*), a C$_4$ plant, and the rest from "nezasa" (*Pleioblastus chino* var. *viridis*), a C$_3$ plant, which may have reflected the fractions of the plant habitat in the grass field in Aso. The radiocarbon analysis showed that LV-WSOC was composed of a substantial amount of modern carbon, which indicated that biomass burning would not discriminate radiocarbon significantly.

**Supplementary Materials:** The following are available online at https://www.mdpi.com/article/10.3390/app12010109/s1, Figure S1: Experimental setup for BAuT recovery tests, Figure S2: Scatter plot of $\Delta^{204}$Hg versus $\Delta^{200}$Hg for TGM from noyaki and background air samples before and after the corrections for the measurement biases were made, Table S1: Summary of noyaki sampling and types of samples collected, Table S2: TGM samples collected from noyaki studies, Table S3: Filter samples collected from noyaki studies, Table S4: Summary of recovery tests through overall procedure using SRM 8610, Table S5: Results of recovery test using IAEA-CH-6 sucrose, Table S6: Comparison between TGM concentrations determined using the conventional traps and BAuTs, Subsection S.1.: TGM recovery test through the overall procedure, Subsection S.2.: Standard spike test of LV-WSOC.

**Funding:** This research and the APC were funded by the National Institute for Minamata Disease (grant ID: RS-20-11 and RS-21-11).

**Institutional Review Board Statement:** Not applicable.

**Informed Consent Statement:** Not applicable.

**Data Availability Statement:** The data presented here are not reported elsewhere.

**Acknowledgments:** The author acknowledges Shoko Science for the $\delta^{13}$C analysis and the Institute for Accelerator Analysis for the $^{14}$C analysis. The author also acknowledges Nanami Yamamoto for her assistance in the laboratory work of this research project.

**Conflicts of Interest:** The authors declare no conflict of interest.

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
