# Peer review of "Isotopic Characterization of Gaseous Mercury and Particulate Water-Soluble Organic Carbon Emitted from Open Grass Field Burning in Aso, Japan"

_applsci, doi:10.3390/app12010109_

Round 1

Reviewer 1 Report

This study entitled “Isotopic Characterization of Gaseous Mercury and Particulate Water-Soluble Organic Carbon Emitted from Open Grass Field Burning in Aso” by Irei reports an interesting study of gaseous mercury and WSOC from open field burning of grass in Aso, Japan. New results are presented in this study and the conclusions appear to be supported by the provided data. Some clarification needs to be made before the manuscript can be published.

Specifics:

Line 21. “elevated” meaning quantitatively?

Line 23. “lighter” meaning?

Line 29. Use “LV-WSOC”

Line 28-30. It is not clear what is the implication from this study other than “an example”.

Line 60-61. Provide a few examples of the isotopic compositions of nascent emissions.

Line 69. What year exactly?

Line 156. “analyzed by” what?

Line 161. How one week was decided for the conversion of gaseous mercury to aqueous Hg2+?

Line 215. Provide more details of how the plant samples were treated. E.g., what weight was used, how they were grounded?

Line 236. If I understand correctly, the biases were not manually applied to the real samples?

Line 245. Blank filters had a value of 10 ugC. Is this blank filter polluted?

Line 272. Is it not possible to do the correction based on the recovery rate?

Table 1. It seems the background air sample on May 10 is very high for total Hg (38 ng m-3) but low for gaseous Hg (1.3 ng m-3). Why is that?

Line 304. Missing caption for Figure 5.

Line 311. “HC” or “HCl”

Line 335. It is not clear how 67% and 33% were estimated.

Techniques:

Line 10. Keywords: Usually 3-5 keywords are sufficient.

Line 314. Remove “between”. Change to “the detection limit of”

Line 346. “demonstrates”

Line 366. Change to “during the Noyaki sampling period”

Reviewer 2 Report

The author presented an isotopic characterization work, originated from open grass field burnings.  Only few comments can be corrected for a better understanding of the manuscript. Moreover, an English revision is suggested.

Line 63: Seeing that the purpose of the research is to characterize also the “low volatile water-soluble nitrogen (LV-WSN)”, then cite it also in the Abstract, next to the LV-WSOC;

Line 110: “iNNOVATION” equalize or invert in capital and lowercase the letters;

Line 113: Add the reference of the “newly developed big gold mercury trap (BAuT)” at this line (Irei, S., 2020), as you mention it for the first time, instead of Line 147;

Line 114-116: Explicit the acronyms of the polymers “PTFE and PFA”;

Line 127-129: There wasn’t a permanent TGM sampling station in the surrouding or regional area, where you could take data and compare them with your measurements?

Line 231: The “(average ± SD, n = 3)” would be better insert it in Line 221-222, when you report the previous data;

Line 304: Add a caption to Figure 5. Moreover, there is no reference to Figure 5 in the manuscript.

Line 311: In “HC” do you mean “HCl”?
